# 'If I am on ART, my new-born baby should be put on treatment immediately': Exploring the acceptability, and appropriateness of Cepheid Xpert HIV-1 Qual assay for early infant diagnosis of HIV in Malawi

Maggie Nyirenda-Nyang'wa[1,2,3]*, Moses Kelly Kumwenda[4,5], Shona Horter[6], Mina C. Hosseinipour[7,8], Maganizo Chagomerana[7,8], Neil Kennedy[9], Derek Fairley[10,11], Kevin Mortimer[3,12,13], Victor Mwapasa[14], Chisomo Msefula[15], Nigel Klein[1], Dagmar Alber[1], Angela Obasi[3,16]

1 Department of Infection, Immunity, Inflammation, Institute of Child Health, University College London, London, United Kingdom, 2 Department of Paediatrics, Kamuzu University of Health Sciences, Blantyre, Malawi, 3 Department of International Public Health, Liverpool School of Tropical Medicine, Liverpool, United Kingdom, 4 Gender in Health Associate Group and Maternal and Fetal Health, Malawi–Liverpool–Wellcome Trust Clinical Research Programme, Blantyre, Malawi, 5 Department of Pathology, Helse Nord TB Initiative, Kamuzu University of Health Sciences, Blantyre, Malawi, 6 Department of Epidemiology and Population Health, London School of Hygiene and Tropical Medicine, London, United Kingdom, 7 University of North Carolina Project–Malawi, Lilongwe, Malawi, 8 Department of Medicine, University of North Carolina at Chapel Hill School of Medicine, Chapel Hill, North Carolina, United States of America, 9 Centre for Medical Education, Queen's University Belfast, Belfast, United Kingdom, 10 Department of microbiology, Belfast Health & Social Care Trust, Belfast, United Kingdom, 11 Wellcome Wolfson institute, Queen's University Belfast, Belfast, United Kingdom, 12 Respiratory Medicine, Aintree University Hospital, Liverpool University Hospitals NHS Foundation, Liverpool, United Kingdom, 13 Department of Medicine, University of Cambridge, Cambridge, United Kingdom, 14 Department of Community and Environmental Health, Kamuzu University of Health Sciences, Blantyre, Malawi, 15 Department of Pathology, Kamuzu University of Health Sciences, Blantyre, Malawi, 16 AXESS Sexual Health, Liverpool University Hospitals NHS Foundation Trust, Liverpool, United Kingdom

* maggie.nyang'wa@ucl.ac.uk

## Abstract

Early infant diagnosis of HIV (EID-HIV) is key to reducing paediatric HIV mortality. Traditional approaches for diagnosing HIV in exposed infants are usually unable to optimally contribute to EID. Point-of-care testing such as Cepheid Xpert HIV-1 Qual assay-1 (XPertHIV) are available and could improve EID-HIV in resource constrained and high HIV burden contexts. We investigated the acceptability and perceived appropriateness of XpertHIV for EID-HIV in Mulanje Hospital, Malawi. Qualitative cross-sectional study using semi-structured interviews (SSI) among caregivers and health care workers at Mulanje District Hospital. The qualitative study was nested within a larger diagnostic study that evaluated the performance of XpertHIV using whole-blood-sample in a resource limited and high burden setting. A total of 65 SSIs were conducted among caregivers (n = 60) and health care providers (n = 5). Data were coded using deductive and inductive approaches while thematic approach was used to analyse data. Point-of-care XPertHIV was perceived to be acceptable among caregivers and health care providers. Caregivers' motivations for accepting XPertHIV HIV-

**Data Availability Statement:** All relevant data are submitted with the manuscript as compressed zip folders with transcripts in Chichewa and translated transcripts in English. These have been attached and made fully available and without restriction as per PLOS Data Policy. The findings are already provided as part of the submitted article.

**Funding:** Maggie Nyirenda-Nyang'wa (MNN) received funding from the following: i) McClay Foundation, 20 Seagoe Industrial Estate, Craigavon, Belfast, Northern Ireland. BT63 5QD via Queens University Belfast - Grant number R2167Cii ii) Fogarty International Center-funded Malawi HIV implementation Research Scientist Training program- Grant number (D43 TW010060). iii) Adalma Foundation via Liverpool School of Tropical Medicine (no grant number or url available) All the funders had no role in study design, data collection and analysis, decision to publish, or preparation of the manuscript.

**Competing interests:** The authors have declared that no competing interests exist.

testing for their infants included perceived risk of HIV emanating from child's exposure and validation of caregiver's own HIV sero-status. Although concerns about pain of testing and blood sample volumes taken from an infant remained amplified, overall, both caregivers and health care providers felt XpertHIV was appropriate because of its quick result turn-around-time which decreased anxiety and stress, the prospect of early treatment initiation and reduction in hospital visits and related costs. Implementation of XpertHIV has a great potential to improve EID-HIV in Malawi because of its quick turn-around-time and associated benefits including overcoming access-related barriers. Scaled implementation of this diagnostic technology require a robust community engagement strategy for managing caregivers and community myths and misconceptions towards the amount of blood sample collected from infants.

## Introduction

High prevalence of HIV-related infant morbidity and mortality have contributed to impeding the attainment of both the UNAIDS 95-95-95 targets and the health-related sustainable development goals [1] Currently, HIV treatment for infants and children lags far behind other demographic groups. Compared to 93% of pregnant women on treatment, only 74% of children aged 0–14 years are receiving antiretroviral therapy (ART) [2]. Early initiation of ART among infants decreases both early infant mortality and HIV progression by approximately 75% [3]. Early diagnosis of HIV in infants (EID-HIV) requires conventional HIV DNA/RNA detection using tests such as polymerase chain reaction (PCR) since maternal antibodies make antibody tests unreliable below 18 months [4,5]. However, HIV PCR is challenging in resource constrained settings because of its high cost, the need for skilled technicians and centralised laboratories [4]. Centralisation of testing also leads to protracted turn-around-time (TAT) with a lag time from testing to receiving results of about 2–3 months. Suboptimal turn-around-times for HIV diagnostic tests has been associated with losses to follow-up of up to 33% [6].

The quality-assured point-of-care test (POCT) Cepheid Xpert HIV-1 Qual assay (whole-blood-sample (XpertHIV)) that can improve EID-HIV is available and has received prequalification for *in vitro* diagnostic use from WHO in 2016 [7,8]. XpertHIV reduces turn-around time to a median of less than one day and is as specific and sensitive as conventional HIV-1 PCR [9,10] and subsequently has the potential for decreasing loss to follow-up and increasing uptake of infant ART through EID-HIV [9].

Malawi, a country that is among those worst affected by HIV, has an estimated 1500 infants who are being diagnosed with HIV every year [6]. Historically, Cepheid GeneXpert diagnostic technologies in the country have been exclusively applied within the context of TB diagnosis through TB assay Xpert MTB/RIF [11,12]. Our recent study at Mulanje District Hospital has demonstrated that XpertHIV had comparable sensitivity and specificity to HIV-1 PCR (Abbott) in diagnosing HIV infection among infants and children but with a much faster turn-around time [9,10]. Similar results have reported in Botswana and Kenya [13,14]. Despite these favourable outcomes of XpertHIV, other factors influence acceptance of using whole-blood sample for EID-HIV such as fears relating to blood sampling, HIV-related stigma, fear of living with a diagnosis of HIV for the infant, difficulties with disclosure to partner or family members, lack of partner support, distance and transport or other costs related to clinic visit for HIV-related services [15–17] This qualitative study investigated healthcare providers and caregivers' perspectives—in terms of acceptability and appropriateness—towards the

introduction of a novel XpertHIV for EID-HIV that use whole-blood sample within a busy rural district hospital.

## Methods

The study took place in Mulanje District Hospital geographically located in the Southern region of Malawi. Mulanje is a mountainous and tea-farming district on a border with Mozambique. The district is predominantly occupied by the Lomwe tribe and has an estimated total population of 428,322 and an adult HIV prevalence 20.6% [18]. Mulanje District Hospital is the secondary referral government hospital for 17 peripheral health centres in Mulanje district. As per the Malawi national HIV guidelines, HIV antibody testing is offered to all HIV exposed children aged >12 months of age. For HIV-exposed infants aged 6 weeks, standard HIV testing uses dried blood spots (DBS) which are sent to Thyolo District Hospital Molecular laboratory (39km away) for HIV-1 PCR testing. From hospital records, approximately 2700 HIV exposed infants from Mulanje district are tested each year.

A diagnostic study was implemented at Mulanje district hospital to evaluate the performance of XpertHIV using whole-blood-sample in a resource limited and high burden context. The performance study comprised a population from which a sample for this qualitative study was drawn. Details of the performance study have published elsewhere [9]. A total of 600 children (aged 0 months– 14 years) were recruited in the feasibility and performance study of XpertHIV of whom 101/600 (17%) were infants aged <1 year. All participants were tested by 5 health providers (i.e. nurses) with both XpertHIV and PCR between July and September 2018 [9,10]. Infant blood was taken by heel prick: 5 x blood spots for PCR (~0.35mls) and 0.5mls into an EDTA for XpertHIV. For ~ 5% of infants where heel prick for capillary blood sampling failed and for older children aged >1 year, venous samples were taken, generally from antecubital veins.

This qualitative cross-sectional study sampled 60 caregivers of children who were recruited in the performance study. Caregivers were purposively sampled, across children's age groups for semi structured interviews (SSI) to ensure maximum variation of the sample. Inclusion criteria for caregivers of HIV exposed infant and children were 1) their decision-making responsibility for an infant or child who was suspected to be HIV exposed but had no prior HIV test; or an HIV exposed infant for EID-HIV; and (2) being over 18 years old; and (3) willing to provide an informed consent. For the health care providers, all five nurses who were working in the outpatient department were purposively selected for SSIs because of their involvement in EID-HIV and prevention of mother-to-child transmission (PMTCT) as well as their role in performing the XpertHIV in the performance study.

Semi-structured interview guides for this study were translated in Chichewa and explored perceptions among implementing nurses and beneficiary caregivers regarding whether or not the use of XpertHIV and DNA PCR is agreeable or satisfactory, including factors influencing acceptance of the test and experiences of EID-HIV. Experiences of interaction with XpertHIV to determine appropriateness and acceptability of this diagnostic technology was also explored among nurses. In terms of appropriateness, our attention was on the perceived fitness, relevance, or compatibility of XpertHIV for EID-HIV in a resource constrained setting. All interviews were conducted in a study room housed within the outpatients' department. Two trained qualitative researchers facilitated the interviews, were responsible for taking field notes, data transcription and translation. Interviews were conducted in Chichewa (dominant local language)—each lasted less than an hour—and were audio recorded. Data collection continued until data saturation was reached [19,20] observable through duplication or repetition of information provided by study participants.

Data were analysed thematically using both inductive and deductive coding to identify emerging themes, patterns, and concepts from participants' accounts. MK developed an analytical coding framework by using the interview guides and listening to the recordings, applying open descriptive coding to transcripts, and considering the focus areas of acceptance, decision-making, and perceptions about XpertHIV. The coding framework was adapted and further developed as analysis progressed. Transcripts were imported into NVivo 11 QSR software (QSR, Melbourne, Australia) for management and coding. Units of emerging themes for analysis were coded from Chichewa transcripts by two Chichewa speakers (MNN and MP) to optimise trustworthiness of interpretation and credibility. Using the coding framework, data was coded by grouping it against the relevant emerging analytical segment. The initial codes were then reviewed for accuracy by the researchers (MK and SH). We triangulated findings from caregivers and nurses to consider how their perspectives converged and diverged. This further enhanced the validity of data. Data has been presented as a descriptive narrative, supported by verbatim quotes. Pseudonyms are used to protect participants' confidentiality.

### Patient involvement

The public, especially mothers and caregivers of infants and children, were involved in some aspects of the study. The mothers and caregivers of children advised that posters about the study should not be put up in the out-patients department to avoid stigma. The pros and cons of different ways of dissemination of information including raising awareness of the study were discussed with them. We reviewed the results with the mothers and caregivers to obtain their perspectives and feedback to ensure that we presented the findings in the most effective way beyond the research community to public.

### Ethics

Ethical clearance was sought from the College of Medicine Research Ethics Committee (COM-REC) of the University of Malawi (P.03/18/2378),The Kamuzu University of Health Sciences and University College London Research Ethics Committee (13313/001). This qualitative study investigated caregivers' and healthcare providers' perspectives hence written informed consent was obtained from the parent/guardian of each participant under 18 years of age and health care providers.

## Results

We interviewed a total of 60 caregivers and all 5 nurses in the outpatient department who were involved in the XpertHIV testing. Of the 60 caregivers, 58 (97%) were female and 57 (95%) were married and had a median age of 25 years (IQR: 22–34) (Table 1). Most caregivers (34/60, 57%) were housewives while some (18/60, 30%) were involved in small scale businesses. Four of the 5 nurses who participated in this qualitative study were female and had a median age of 33 years (IQR: 23–58) (Table 1).

Five main themes emerged from analysis of caregiver and nurse perspectives on the acceptability and appropriateness of XpertHIV at Mulanje district hospital. These have been divided into factors influencing XpertHIV EID-HIV acceptance and views about the appropriateness of this technology within a context of a national roll-out.

### Factors influencing acceptability of XpertHIV for EID-HIV

To explore caregivers and healthcare providers acceptability of XpertHIV, we looked at factors that influenced end-user acceptance of XpertHIV for EID-HIV. Emerging themes have been

**Table 1. Demographic characteristics of study participants.**

| Characteristic | Frequency | |
|---|---|---|
| | **Caregivers n = 60** | **Nurses n = 5** |
| **Age (median)** | 25yrs (IQR: 22–34) | 33 yrs (IQR: 23–58) |
| **Sex** | | |
| Male | 2 (3%) | 1 (20%) |
| Female | 58 (97%) | 4 (80%) |
| **Marital Status** | | |
| Married | 57 (95%) | 4 (80%) |
| Separated/Single | 3 (5%) | 1 (20%) |
| **Occupation** | | |
| Working in a farm | 7 (11%) | NA |
| Working in small scale business | 18 (30%) | |
| Unemployed/Housewife | 34 (2%) | |
| Civil Servant | 1 (57%) | |

IQR: Interquartile range NA: Not applicable.

presented in five groups namely caregivers' motivations for accepting XpertHIV testing for EID-HIV; Trust in providers professionalism; Gendered engagements with XpertHIV and satisfactory turn-around-time. However, caregivers expressed certain fears toward XpertHIV procedures which in effect negatively affected the acceptability of the technology for EID-HIV.

**(a) Perceived benefits of EID-HIV.** Caregivers were very protective of their young children and accepted their infants to undergo HIV testing using XpertHIV because of the perceived benefit of ensuring early HIV diagnosis and access to early treatment for the young children. They considered access to early treatment as valuable for a prospect of improved and better lifetime health. Caregivers agreed that their children needed ART treatment initiation immediately after diagnosis because to do so meant safeguarding current and future health and wellbeing of the child as illustrated in a quote below.

> ". . .if I am on antiretroviral therapy, my new-born child should receive ART as soon as possible."

[Caregiver 7, 43-year-old woman, child aged ~3 years]

Most caregivers found EID-HIV satisfactory mainly because of their desire to have an expeditious HIV diagnosis of their children and to ensure that their children have a healthy life. Caregivers were very keen to access HIV testing for their young children to enable them to explore the best available options for prompt access to relevant healthcare follow-up after a positive result. A caregiver cited in a quote below was emphatic about prompt treatment of the child.

> ". . . using this method we can know the status of the child and how we can take care of him/her, access treatment if found infected, and if found negative how we can prevent."

[Caregiver 1, 25-year-old woman, child aged ~1 year]

The obvious risk of the child contracting HIV from their mother living with HIV appeared as a strong incentive that drove caregivers to accept XpertHIV test despite their lack of

understanding about potential routes of HIV acquisition among children but also the apparent prevailing denial/othering of potential sources of HIV infection in infants and young children:

*". . .. children should be tested because children might have contracted HIV somewhere without their parents knowing. Because they play different games and with a wide range of objects."*

[Caregiver 63, 25-year-old woman, child aged ~1 year]

In a quote above, a mother living with HIV valued a prompt HIV test for her child through XpertHIV but removed the potential source of HIV infection for her infant from herself. This was possibly because of denial of the guilt of exposing an innocent child to HIV or mechanism for coping with a potential HIV-positive result for the child.

Some caregivers also accepted their children to undergo a technologically advanced EID-HIV using XpertHIV as a tool for verifying own HIV status by proxy. In this context, caregivers seemed to distrust the routine lateral flow antibody test that are used in Malawi for HIV diagnosis in children aged >1 year. As such, some caregivers who had a positive test-result in their health passport described having tested negative previously and raised concerns about the possibility of them being in the window period when they were initially tested. They therefore accepted EID-HIV simply to confirm their HIV status. Thus, testing children generally was described by caregivers who distrusted previous HIV-positive results as a means through which own HIV status was ascertained:

*"Start with testing children because from there it is easy for adults to know their status."*

[Caregiver 20, 25-year-old woman, child aged ~4 years]

**(b) Professionalism of health providers.** Trust in the work and professionalism of nurses among caregivers was an important factor facilitating acceptance XpertHIV for EID-HIV at Mulanje district hospital. Trust in health care providers was linked to their perceived superior knowledge in HIV matters; perceived expertise among nurses in HIV diagnosis including conducting a venipuncture procedure in infants, and a sense of dependence on nurses as the gatekeepers to health services. It appeared to be supported by friendly communication and supportive practitioner-patient relationships. Several caregivers also described the importance of confidentiality, trusting that the nurses would maintain their privacy and look after their best interests.

*"People have faith in health workers because they [health workers] are the ones who provide help when someone is sick".*

[Caregiver 68, 33-year-old woman, child aged 6 years]

*"Because the hospital staff know everything so they must be trusted."*

[Caregiver 62, 40-year-old woman, child aged 3 months]

The assumption that the second quote is making is that health care providers are equipped with sufficient knowledge and skills for providing health care that is in the interest of the patients. A complete trust in the ability of health care providers meant that the caregivers didn't have sufficient power or authority to ask for more information in order to fully comprehend XpertHIV test for EID-HIV.

**(c) Gendered decisions-making.**   Study findings demonstrated gendered dimensions embedded within decision-making to accept or decline participation of a child in XpertHIV testing for EID-HIV. Most caregivers' accounts alluded to men's position of influence within the household, as men held enormous amount of power for authorising HIV testing for any member of their household including the infant. During data collection, some health care providers observed that women caregivers were often describing the difficulties they experienced in discussing subjects pertaining to HIV testing services without consulting or seeking approval from their male partners. Making important decisions without the involvement of a male partner was viewed as a deviation from cultural norms as captured in a quote below:

*"I also feel it is because of the culture that says that decision making belongs to the man, and this causes [women] not to answer questions."*

[nurse03 male, single, 23 years]

Women caregivers felt that because of the social and household position that men hold, it is important to encourage them get involved in health-related matters including processes of testing for HIV among their children. Infusing in men the need to be involved in health-related matters is important to ensure that they shoulder the mantle of health issues and make progressive decisions including motivating women in health-related matters as shown in this quote:

*"Men should be coming to the hospital with their families to get tested as the heads of the families and we should motivate them as women".*

[Caregiver 37, 40-year-old woman, child aged 5 years]

**(d) Rapid turn-around-time of XpertHIV.**   The availability of same-day results was recognised by both the caregivers and health care providers as an important feature and benefit of EID-HIV using XpertHIV which is absent in the available routine approaches. Study participants argued that same-day results were essential in permitting speedy access to treatment, care and support for an HIV infected child. Caregivers were concerned that delayed initiation of treatment for HIV infected infants was likely to result into undesirable health outcomes.

*"[I want] Same-day, because you are waiting to hear how your child will be, so it's important to know fast, to see how you will help the child."*

[Caregiver 4, 20-year-old woman, child aged ~2 years]

Same-day results also meant that caregivers were not living in suspense for prolonged periods before knowing about HIV status of their child. Waiting for test results for extended period of time and the linked uncertainties surrounding thus was said to be associated with stress, worry, and anxiety on the part of the caregiver. Caregivers felt that the long waiting time for test results was inconsistent with the expected turn-around time (TAT) of most routine tests and had the potential to undermine future acceptance of the test.

*"When we come for testing, we need to hear the results on the same day because we have worries about the possible results."*

[Caregiver 1, 25-year-old woman, child aged ~1 year]

An optimal turn-around-time of results through XpertHIV was thus considered to be consistent with the expectations of caregivers of receiving prompt health care when needed. Additionally, caregivers also described the risk of forgetting to return for test results after time has passed, which was said to influence disengagement from the EID-HIV cascade.

*"Same day [is better], because when you get tested and time passes by you might forget."*

[Caregiver 53, 24-year-old woman, child aged ~2 years]

Nurses' accounts echoed caregivers' perspectives as they valued the quick turn-around time for results with XpertHIV. Apart from the benefit of prompt results given to the caregivers, health care providers felt that XpertHIV was advantageous because it supported them in performing their work. Use of this technology meant that caregivers were providing health care to children without delays on the basis of unavailable test results which are important for decision-making on whether to initiate treatment or not:

*"I feel good because this method is fast, and it will help us to save lives in time."*

[Nurse01 female, married, 33 years].

**(e) Caregivers fears and anxieties towards XpertHIV EID-HIV.** Several caregivers expressed the importance of counselling for the caregiver to come alongside EID-HIV to address and provide support around service user's fears surrounding the testing process. In this context, the perceived 'counselling' was often synonymous with information giving about XpertHIV. Providing sufficient information to caregivers about what they should expect with regard to the process that their child would undergo was said to be vital to ensure that they were sufficiently emotionally prepared and ready to undergo this process and for HIV results:

*"Doctors should offer counselling to reduce the person's stress when waiting for the results."*

[Caregiver45, 38-year-old woman, child aged 9 years]

Caregivers expressed some negative perceptions that had the potential of undermining XpertHIV test acceptance for EID-HIV. These included concerns about process of drawing a large quantity of blood from a child and a potential physical harm/injury. Interestingly, the process of blood drawing using capillary sampling for XpertHIV EID-HIV test was comparable to that of DBS used for routine EID-HIV. The main concerns of caregivers were primarily centred on the quantity of blood that was drawn from an infant; the physical pain to the child from an intrusive procedure and the experience of recurrent struggles by nurses just to collect a blood sample. It is important to underline that the parent performance study demonstrated that nurses experienced difficulties in drawing blood sample from about 5% of children:

*"My concern is on the vein which blood is taken from, which is painful to a child."*

[Caregiver 11, 30-year-old woman, child aged 4 years]

*". . . my fear is the venous blood drawing. I am afraid of it leaving a wound on the puncture site."*

[Caregiver48, 36-year-old woman, child aged 4 years]

The first quote is about the pain that the child had to endure during XpertHIV testing while the second quote speaks of the potential for irreparable physical injury to the child. The nurses agreed that they sometimes experienced difficulties when performing venepuncture procedures to draw venous blood samples from infants and young children. They frequently described blood drawing as one of the key challenges of EID-HIV using XpertHIV testing procedures.

*"But it EID-HIV is a challenge when it comes to finding a vein in children under 6 years."*

[Nurse05, female, married, 28 years].

As expected for Mulanje and the Malawi context broadly, some caregivers had great concerns about what happened to the infant's blood once it was drawn. Care givers seemed to be uncertain and worried about the additional use of the blood or procedures that ensued after the HIV test. These fears seemed justified considering that caregivers had previously tested using the routine rapid antibody test that uses a small sample of blood to make an HIV diagnosis. To the caregivers, the quantity of blood samples that had been drawn from their infants and children was felt to be too much for a simple test. They seemed to feel that there was an added sinister use of the blood sample as shown in this quote:

*"I have great concern with where the blood goes after being taken."*

[Caregiver06, 30-year-old woman, child aged 9 years]

*"My problem is with the amount of blood taken and what they do with it."*

[Caregiver56, 25-year-old woman, child aged 9 years]

Fears expressed in the second quote emerged largely because most of the caregivers had prior knowledge and experience of the amount of blood needed for an HIV test using the routine rapid antibody tests in adults.

## Appropriateness of XpertHIV EID-HIV in scaled contexts

Collectively, the caregivers and HCWs considered XpertHIV as appropriate for Malawi and proposed the need for scaling-up its availability even at a primary health facility level. Caregivers especially felt that making XpertHIV testing available at the primary health facility level was likely to be received with great appreciation by community members. They cited several potential benefits of scaling-up the implementation of XpertHIV testing in the primary health facilities and rolling-out this programme nationally such as increased access to HIV services and shorter walking distances to accessing HIV tests for infants.

*"We would be happy because the HIV testing service has come closer to us, unlike accessing it from a long distance."*

[Caregiver04, 20-year-old woman, child aged ~2 years]

Caregivers held varied and sometimes divergent perspectives regarding affordability and scalability of XpertHIV. Some felt the Ministry of Health (MoH) would be very interested in this technology on the account of excellent turn-around- time of results and because it can be operated by non-lab healthcare providers. They also indicated that scaled implementation of XpertHIV could reduce the existing burden and linked logistical problems associated with the available approaches for providing EID-HIV:

*"The ministry of health will be interested because the whole process is very fast"*

[Nurse01 female, married, 33 years]

Other nurses considered availability of resources as a key determinant of whether the government could afford to scale up XpertHIV countrywide and sustain implementation in health facilities. This group of health care providers recognised inadequate financial and human resources as key potential barriers to implementation of the XpertHIV testing approaches to scale. They also expressed that Malawi as a country was already struggling financially in providing adequate support to health facilities as demonstrated through current and previous experiences while working at the government health establishments.

*"Considering the financial difficulties facing the country today, I think it would be hard for the government to purchase equipment."*

[Nurse01 female, married, 33 years].

Despite these potential barriers to the implementation of XpertHIV, healthcare providers highly valued the use of XpertHIV specifically citing ease of use to providers and timely results as key selling points for this technology. Usability data collected from 7 laboratory technicians were favourable and showed XpertHIV as an appropriate technology for this context. All the laboratory technicians found the XpertHIV user-friendly, had a quick TAT of results compared to the existing EID-HIV testing approaches, and did not experience power supply problem during implementation. Most of the laboratory technicians (5/7) narrated that the Cepheid GeneXpert instrument and XpertHIV assay was easy to use and all equipment together with reagents were readily available in Malawi. Most of the laboratory technicians [5/7] expressed a view that XpertHIV should be adopted at Mulanje District Hospital. Only 2 laboratory technicians felt that the introduction of XpertHIV at their hospital would increase their workload and preferred sending samples to another facility for testing. Considering the ease of use of XpertHIV, all laboratory technicians agreed that XpertHIV should be adopted and deployed in district hospitals all across Malawi. They maintained that scaling up XpertHIV would not be very difficult for the government to achieve since GeneXpert equipments are already available in most laboratories of most district hospitals in Malawi. However, this move may require procurement of additional Cepheid GeneXpert equipments for use in facilities where these equipments are non-existent or to replace non-functional machines. It would also require speedy training for both new and existing health personnel on usage and maintenance of this technology.

*"Yes, we need a lot of time for orientation because someone new needs to be properly trained."*

[Nurse01 female, married, 33 years].

## Discussion

This study that explored perspectives towards the introduction of XpertHIV that uses whole blood sample for EID-HIV has qualitatively demonstrated acceptance of XpertHIV for EID-HIV. Caregivers accepted EID-HIV using XpertHIV mainly because of their desire to protect their young children from HIV through early diagnosis and subsequent prompt access to ART. In this case, testing through XpertHIV was employed as a gateway to knowledge of a child's status, treatment, and support. These findings are similar to what another study from

Lesotho reported that early knowledge of children's HIV status was the primary motivator for caregivers to accept EID-HIV [21]. In the same way, some caregivers in this study also accepted EID-HIV for their children to confirm their own HIV status by proxy, reverberating similar findings from South Africa [17].

Trust, as shown by this study, is essential for acceptance of health interventions or other therapeutic encounters by caregivers which was also reported elsewhere [22]. For example, findings from research on cancer has highlighted how trust shapes patients' engagement with treatment and health care services [23]. While we found that trust in healthcare providers professionalism and skills to collect whole-blood sample appeared to positively shape caregivers' acceptance and engagement with EID-HIV using XpertHIV, it would be important to critically examine instances where practitioner-patient relationships are not perceived as positively as has been described in other settings [24]. Whilst blood sampling with its associated pain and fear of scarring was of concern by some caregivers in this study, and the amount of blood drawn from an infant and worry about potential clandestine use was another major concern. This finding signifies a disconnect between trust in practitioners and trust in the system by some caregivers, as the routine DBS and XpertHIV assays require blood sampling and if capillary sampling fails, venous sampling is done therefore this will not be unique to XpertHIV.

Despite these anxieties, XpertHIV testing using whole-blood-sample was still acceptable to caregivers and healthcare providers since they perceived the benefits of knowing the HIV status of the child to outweigh the underlined concerns.

Healthcare providers and caregivers' experiences with a novel XpertHIV for EID-HIV that uses whole-blood-sample was followed with perceptions that the use of this diagnostic technology was agreeable and satisfactory. The study has also shown that the introduction of XpertHIV that uses whole-blood-sample was perceived to be relevant and compatible to addressing the current challenges that the hospital experiences with EID-HIV.

Interestingly, acceptance of XpertHIV EID-HIV using whole-blood-sample by caregivers was shown to be influenced by household power and gender dynamics as male partners involvement was key in decision-making processes around uptake. Within the Malawian context, men are generally viewed as primary decision-makers for the household including making decisions about the health of children such as HIV testing. Thus, the lack of engagement of men in health programmes and interventions can be a strong barrier to uptake of health services by other family members. Kumwenda and colleagues [25] for example, reported that men were more likely than women to fear testing for HIV with their partner using a novel HIV test because a positive result had the potential to expose their status and subsequently render them vulnerable to blame and accusation for introducing HIV into their relationships [25]. The position of influence for men at household level can enable or disempower caregivers, usually women, to seek EID-HIV for their children. A study in Kenya described supportive roles men can adopt accordant with cultural conceptions of masculinity, which can still facilitate infants' uptake of EID-HIV [26]. There is a need to further investigate the role of household gender and power dynamics in decision making processes on health-related matters and identify optimal approaches for engaging male partners in HIV diagnostic services for their children using innovative and technologically advanced strategies such as XpertHIV.

A quick TAT using XpertHIV that use a whole-blood-sample was shown to ease some of caregivers' anxieties linked to long waiting time for test results and problems associated with making follow-up. Healthcare providers also found XpertHIV using a whole-blood-sample to address logistical nightmares linked to transportation of samples to a secondary testing site. Other settings have also reported elevated levels of caregivers' anxiety while waiting for test results of their children, which are exacerbated by mothers' feelings of responsibility for the

test outcome, including guilt related to the chance of an innocent child testing positive for HIV [16].

Although caregivers perceived XpertHIV using a whole-blood-sample as appropriate for their district hospital, they held varied and sometimes divergent views about affordability and scalability or national roll-out of XpertHIV based on their understanding of the infrastructural and cost implications of scaled implementation. Through their experiences in working within the Malawian health system, health providers expressed concerned about resource availability and supply chain to support scaled implementation of XpertHIV using a whole-blood-sample. This is primarily because Malawi is among the poorest countries, ranking 164th out of 177 on the UNDP Human Development Index (HDI) [27] and her economic progress has stagnated since 1990 largely due to the negative impact of HIV/AIDS, poverty and illiteracy [27]. Having such a fragile economy makes it difficult for the country to financially support scaled implementation of health interventions that are costly. However concerns regarding battery power back up associated with the XpertHIV testing were also raised by other health workers in other studies [13].

The strength of this study was that it was conducted in a context where XpertHIV using a whole-blood-sample was novel to both nurses and caregivers, thereby offering an opportunity to explore perspectives on its acceptability and appropriateness. These findings may change overtime as the XpertHIV tests for EID-HIV using a whole-blood-sample becomes more available. The sample size of 60 caregivers was sufficient because the study attained saturation of information noticeable through repetition of themes, and thus providing the confidence that the findings were robust. Furthermore, the high proportion of mothers included as caregivers was representative of the clinical context where women are observed to take a leading role in providing support to family members when ill.

The study included only 5 healthcare providers who were all nurses who were involved in the parent study. The insights provided by these nurses were sufficient for the purposes of this study. Caregivers may have associated the interviewers with the health workers who were responsible for provision of health services, which could have influenced their accounts however we probed more to gather the most narratives.

## Conclusions

This study has provided useful insights about the introduction and implementation of a novel XpertHIV using a whole-blood-sample through caregivers' and healthcare providers' experiences following their engagement with this technology.

EID-HIV through XpertHIV that use a whole-blood-sample was acceptable among both caregivers' and healthcare providers for a variety of reasons including protecting the health of a child and confirming one's HIV status by proxy. Future scaled implementation of EID-HIV through XpertHIV using a whole-blood-sample would require addressing the barriers to uptake including pre-existing fears about blood taking and trust in the use of blood samples. The caregivers emphasised the need for pre and post counselling to prepare them for XpertHIV testing, acceptance of test results and linkage to HIV care if their infants and children are diagnosed with HIV infection.

XpertHIV was also deemed appropriate for a rural district hospital among both caregivers' and healthcare providers due to its prompt turn-around-time of results when compared to the existing diagnostic approaches, simplicity and ease of use by less skilled health providers. Healthcare providers made strong recommendations for scaled implementation across the country because of these benefits. XpertHIV has great potential for improving early HIV diagnosis among infants in resource constrained settings however prior to national roll-out of

XpertHIV, resource and supply implications should be considered to ensure XpertHIV access is universal and sustainable.

## Recommendation

This intervention is important for scale up however governments, policy makers and programme planners should have information about caregivers' and healthcare providers' experiences and engagement with XpertHIV using a whole-blood-sample and why it's necessary to improve TAT as per findings of this qualitative study. There should also be enough heath workers in paediatric outpatient setting and wards where HIV testing of infants and children occurs.

Since this was a process evaluation of implementation research on XpertHIV using whole blood, we made recommendations in the main paper.

More implementation studies to assess engagement with XpertHIV in other sites nationally and in the region should be conducted whilst scaling-up this intervention.

## Supporting information

**S1 Table. Codebook.**
(XLSX)

**S1 Text. Interview guide for caregivers.**
(DOCX)

**S2 Text. Interview guide for health workers.**
(DOCX)

**S1 File. Caregivers' transcripts.**
(ZIP)

**S2 File. Health workers and caregivers' transcripts.**
(ZIP)

## Acknowledgments

The authors thank the participants and their caregivers for consenting to participate in this study. We also acknowledge the contribution of the staff of Mulanje District Hospital especially the laboratory staff and Deputy District Hospital Officer; HIV DETECT study team, paediatric nurses and health surveillance assistants for their support in recruitment of participants; the qualitative researchers/transcribers; and the Ministry of Health in particular Mr James Kandulu, the Acting Director of Diagnostics.

## Author Contributions

**Conceptualization:** Maggie Nyirenda-Nyang'wa, Dagmar Alber.

**Data curation:** Maggie Nyirenda-Nyang'wa.

**Formal analysis:** Maggie Nyirenda-Nyang'wa, Moses Kelly Kumwenda, Shona Horter.

**Funding acquisition:** Maggie Nyirenda-Nyang'wa, Mina C. Hosseinipour, Neil Kennedy, Derek Fairley, Victor Mwapasa.

**Investigation:** Maggie Nyirenda-Nyang'wa.

**Methodology:** Maggie Nyirenda-Nyang'wa, Moses Kelly Kumwenda.

**Project administration:** Maggie Nyirenda-Nyang'wa.

**Resources:** Maggie Nyirenda-Nyang'wa.

**Software:** Maggie Nyirenda-Nyang'wa.

**Supervision:** Maggie Nyirenda-Nyang'wa, Moses Kelly Kumwenda, Mina C. Hosseinipour, Chisomo Msefula, Nigel Klein, Dagmar Alber, Angela Obasi.

**Validation:** Maggie Nyirenda-Nyang'wa.

**Visualization:** Maggie Nyirenda-Nyang'wa.

**Writing – original draft:** Maggie Nyirenda-Nyang'wa, Moses Kelly Kumwenda, Shona Horter.

**Writing – review & editing:** Moses Kelly Kumwenda, Shona Horter, Mina C. Hosseinipour, Maganizo Chagomerana, Neil Kennedy, Derek Fairley, Kevin Mortimer, Victor Mwapasa, Chisomo Msefula, Nigel Klein, Dagmar Alber, Angela Obasi.

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
