## [Decision Letter · Decision Letter 0]

22 Jul 2022

PGPH-D-22-00882

‘If I am on ART, my new-born baby should be put on treatment immediately’: Exploring the a cceptability , and appropriateness of Cepheid Xpert HIV-1 Qual assay for early infant diagnosis of HIV in Malawi.

Dear Dr. -Nyang'wa,

Thank you for submitting your manuscript to PLOS Global Public Health. After careful consideration, we feel that it has merit but does not fully meet PLOS Global Public Health’s publication criteria as it currently stands. Therefore, we invite you to submit a revised version of the manuscript that addresses the points raised during the review process.

We look forward to receiving your revised manuscript.

Kind regards,

Elisa Lopez-Varela, MD, MPH, PhD

Academic Editor

Journal Requirements:

a. Please clarify all sources of funding (financial or material support) for your study. List the grants (with grant number) or organizations (with url) that supported your study, including funding received from your institution. 

b. State the initials, alongside each funding source, of each author to receive each grant.

c. State what role the funders took in the study. If the funders had no role in your study, please state: “The funders had no role in study design, data collection and analysis, decision to publish, or preparation of the manuscript.”

2. Please ensure that the Title in your manuscript file and the Title provided in your online submission form are the same.

Additional Editor Comments (if provided):

Reviewers' comments:

Reviewer's Responses to Questions

**Comments to the Author**

1. Does this manuscript meet PLOS Global Public Health’s publication criteria? Is the manuscript technically sound, and do the data support the conclusions? The manuscript must describe methodologically and ethically rigorous research with conclusions that are appropriately drawn based on the data presented.

Reviewer #1: Yes

Reviewer #2: Yes

2. Has the statistical analysis been performed appropriately and rigorously?

Reviewer #1: N/A

Reviewer #2: N/A

3. Have the authors made all data underlying the findings in their manuscript fully available (please refer to the Data Availability Statement at the start of the manuscript PDF file)?

Reviewer #1: Yes

Reviewer #2: No

4. Is the manuscript presented in an intelligible fashion and written in standard English?

Reviewer #1: Yes

Reviewer #2: Yes

5. Review Comments to the Author

Reviewer #1: The authors provide a detailed description of the acceptability and appropriateness of XpertHIV diagnostic testing in infants and young children in rural Malawi. The manuscript is clear, well written and provides a valuable contribution to the field. I have added only minor edits and comments.

Abstract:

Word missing: “The qualitative [study/analysis/component] was nested within a larger diagnostic study…”

“Data was coded..” Should be data were coded

Background

Page 3: Authors mention internalised stigma but could also relate to other forms of stigma – anticipated stigma.

Line 101 – 104: The authors mention a list of patient/caregiver centred fears/concerns. Are there any studies/data related to health worker concerns? For instance resources, training, work load etc.

Line 142 – 147: Consider moving this paragraph and the table into the results/findings section.

Table 1: Domestic work and housewife are grouped together. Should these not be reported as distinct categories? Is this income generating domestic work?

Methods: How did discussion guides differ between health workers and caregivers?

Ethics:

A minor point: In the inclusion section, the authors note that participants had to be either over 18 years old or emancipated minors. In the ethics sections, the authors note that informed consent was obtained from the parent/guardian of each participant under 18 years of age. This seems to be contradictory – either participants are over 18 or emancipated, so no consent from guardians would be required.

Results:

It would be interesting to note how the diagnostics process was described and explained to caregivers. Understanding of the test and knowledge of HIV and risk to infants could all impact the acceptability of the diagnostic test, as the authors briefly note in lines 271 – 274.

Line 316: “Genderred” should be ‘gendered’

Line 332: Were there any mention of men refusing to have children tested for HIV?

Line 362: Write out TAT first time used

Line 433: This is an interesting finding and has the potential to act as barrier to future uptake. Did any of the caregivers mention what they think happened to the blood (sinister use)?

Conclusion

The authors made note in the methods section how participants provided input on how to convey information on the diagnostic test. There is scope to include how the caregivers would like the information to be shared in future. What should be emphasised and how should caregivers be informed/prepared? The authors can consider expanding on this in the conclusion section.

Reviewer #2: To the authors:

Congratulations on the manuscript, very concerning topic and a good piece of evidence. Congrats!

As qualitative researcher, I really enjoyed reading your lines.

See my comments below.

Line 43 – I suggest using EID-HIV FOR Early infant diagnosis of HIV

Line 196 – I suggest sharing the coding book for didactic purposes since the article is likely to be read by a variety of people including beginners in qualitative research analysis.

Line 219 – please remove the “now” in the sentence.

The Ethics statement is incomplete regarding the mothers and caregivers as well as the health care providers please include.

Line 465 – “nurse” is lowercase,

Line 474 – “nurse” is sentence case

Please revise and standardize the presentation of quotes and participants in brackets.

Line 585 – I do not see the sample of nurses as a limitation since it was very well justified why this number.

I would suggest the authors to include a recommendation sections since there is a possibility of scale-up of the intervention. The recommendations would be useful to other sites nationally to consider and improve the implementation.

All the best with your manuscript.

6. PLOS authors have the option to publish the peer review history of their article (what does this mean?). If published, this will include your full peer review and any attached files.

**Do you want your identity to be public for this peer review?** For information about this choice, including consent withdrawal, please see our Privacy Policy.

Reviewer #1: No

Reviewer #2: No

---

## [Decision Letter · Decision Letter 1]

6 Dec 2022

PGPH-D-22-00882R1

‘If I am on ART, my new-born baby should be put on treatment immediately’: Exploring the acceptability , and appropriateness of Cepheid Xpert HIV-1 Qual assay for early infant diagnosis of HIV in Malawi.

Dear Dr. Nyirenda -Nyang'wa,

Thank you for submitting your manuscript to PLOS Global Public Health. After careful consideration, we feel that it has merit but does not fully meet PLOS Global Public Health’s publication criteria as it currently stands. Therefore, we invite you to submit a revised version of the manuscript that addresses the points raised during the review process.

Please address the remaining minor considerations provided by Reviewer 1. 

We look forward to receiving your revised manuscript.

Kind regards,

Vanessa Carels

Staff Editor

Journal Requirements:

Additional Editor Comments (if provided):

Reviewers' comments:

Reviewer's Responses to Questions

**Comments to the Author**

1. If the authors have adequately addressed your comments raised in a previous round of review and you feel that this manuscript is now acceptable for publication, you may indicate that here to bypass the “Comments to the Author” section, enter your conflict of interest statement in the “Confidential to Editor” section, and submit your "Accept" recommendation.

Reviewer #1: All comments have been addressed

Reviewer #2: All comments have been addressed

2. Does this manuscript meet PLOS Global Public Health’s publication criteria? Is the manuscript technically sound, and do the data support the conclusions? The manuscript must describe methodologically and ethically rigorous research with conclusions that are appropriately drawn based on the data presented.

Reviewer #1: Yes

Reviewer #2: Yes

3. Has the statistical analysis been performed appropriately and rigorously?

Reviewer #1: N/A

Reviewer #2: N/A

4. Have the authors made all data underlying the findings in their manuscript fully available (please refer to the Data Availability Statement at the start of the manuscript PDF file)?

Reviewer #1: Yes

Reviewer #2: Yes

5. Is the manuscript presented in an intelligible fashion and written in standard English?

Reviewer #1: Yes

Reviewer #2: Yes

6. Review Comments to the Author

Reviewer #1: The authors have addressed all of the comments. A few (very minor) suggested edits:

Throughout the text, please refer to mothers or children “living with HIV rather than “HIV-positive”.

Line 232: "Written informed consent was obtained from the parent/guardian of each participant under 18 years of age and health care providers.. " Does participants refer to the children or the caregivers? Or does it refer to caregivers/mothers who are under the age of 18? If so, did they complete assent as well?

Line 350 typo: "approval from their male partner’s" - should be "male partners"

Reviewer #2: Comments on manuscript: ‘If I am on ART, my new-born baby should be put on treatment immediately’: Exploring the acceptability, and appropriateness of Cepheid Xpert HIV-1 Qual assay for early infant diagnosis of HIV in Malawi.

Revision 1 – November 22

To the editor:

Thank you for the opportunity to review the manuscript.

Regards.

To the authors:

Congratulations again on the revised manuscript!

I delighted to see that the comments and suggestions from my first review were agreed by the authors and included.

I have no further comments.

All the best with your manuscript.

7. PLOS authors have the option to publish the peer review history of their article (what does this mean?). If published, this will include your full peer review and any attached files.

**Do you want your identity to be public for this peer review?** For information about this choice, including consent withdrawal, please see our Privacy Policy.

Reviewer #1: No

Reviewer #2: **Yes: **Neusa Torres

---

## [Editor Report · Decision Letter 2]

17 Jan 2023

‘If I am on ART, my new-born baby should be put on treatment immediately’: Exploring the acceptability, and appropriateness of Cepheid Xpert HIV-1 Qual assay for early infant diagnosis of HIV in Malawi.

PGPH-D-22-00882R2

Dear Dr -Nyang'wa,

We are pleased to inform you that your manuscript '‘If I am on ART, my new-born baby should be put on treatment immediately’: Exploring the acceptability, and appropriateness of Cepheid Xpert HIV-1 Qual assay for early infant diagnosis of HIV in Malawi.' has been provisionally accepted for publication in PLOS Global Public Health.

Best regards,

Julia Robinson

Executive Editor